# The Hegemonic Character of Techno-Functional Neo-Immanentism and Its Relationship with Culture Wars

Celso Sánchez Capdequí, Javier Gil-Gimeno and Pablo Echeverría Esparza *

I-Communitas, Institute for Advanced Social Research, Public University of Navarra, 31006 Pamplona, Spain;
celso.sanchez@unavarra.es (C.S.C.); fcojavier.gil@unavarra.es (J.G.-G.)
* Correspondence: pablo.echeverria@unavarra.es

**Abstract:** This paper analyzes the social processes that have led to the consolidation of a technocratic secular order and the type of cultural struggle that has made this possible. To this end, it first proposes a reconstruction of the technocratic consciousness in the course of the secularization process that culminates in the *technological determinism* or *technological solutionism* of the social present; then, the analysis focuses on the *neo-immanentist* tendency of techno-functionalism, in which the secular context and the text of secularization become *one* and deplete a social explanation; thirdly, it reflects on and deals with the open nature of secular life, in which context does not determine social texts (inter-actions) and opens the way to the existence of different *life options* that compete with each other and even turn on—rebel against—institutional design. This reflection, then, focuses on the specific features of the culture wars in Western Judeo-Christian culture and its globalizing tendency. Finally, the document closes with a conclusion that analyzes the road travelled and introduces the new challenges arising from the arguments presented.

**Keywords:** neo-immanentism; technological determinism; religious; secular; technological solutionism; culture wars

## 1. Introduction

In the social context of our time, culture wars define the contours of public debate. They give rise to opposing, and sometimes acrimonious, positions on issues of enormous significance for cultural narratives that struggle to occupy the symbolic centrality of society. Global warming, multiculturalism, euthanasia, immigration, and gender violence, among others, constitute the relevant debates on the public agenda and force their actors to take positions and defend them. However, these debates are not merely a matter of adjudicating or withdrawing reason in the manner of a dispassionate academic debate. The different cultural positions, as James D. Hunter says, involve "commitments and beliefs that provide a source of identity, purpose and togetherness for the people who live by them. It is for precisely for this reason that political action rooted in these principles and ideals tends to be so passionate" (Hunter 1991, p. 42). These intense struggles that take place in the public domain express cultural positions that struggle for moral authority, which is understood as "the basis by which people determine whether something is good or bad, right or wrong, acceptable or unacceptable" (Hunter 1991, p. 42).

In this sense, it is of utmost importance to underline that the condition of possibility of these culture wars—at least in the scenario of the modern Western Judeo-Christian cultural program[1], which is the context in which this work will be developed—is the existence of a secular context that promotes diversity, debate, and the right to dissent as mechanisms of social coexistence. In a way, the debates deployed in terms of cultural warfare have typically modern origins, features, and developments. That is to say, "both the battlefield—which is none other than the civil and democratic sphere—and the objectives pursued—recognition in terms of normativity of their claims—have a markedly modern imprint—that is, they

can only be explained in the scenario of societies in which there has been a recognition of civil rights for society as a whole and in which these rights have been extended to other fields such as the political and social, as stated by Thomas H. Marshall" (Aguiluz et al. 2022, p. 91). The globalized social context, called the *immanent frame* by Charles Taylor (2007), generates the conditions of possibility in which different cultural sensibilities confront each other's moral positions and seek the adhesion of the rest in order to reinforce themselves to the detriment of their adversaries. In a way, this secular scenario defined by science, the market, and information (Taylor 2007) corresponds to democratic social models in which social sensibilities must position themselves and explain themselves to others in order to gain social legitimacy in the face of a specific problem. In it, believers, atheists, scientists, feminists, ecologists, and many other sensibilities speak and collide in processes of dialectical confrontation with always-unpredictable results. *Nothing is written or in the hands of a single voice or authority*. The *immanent frame* is elastic enough to assume different patterns resulting from cultural struggles. Furthermore, as a result of these struggles, its legal, regulatory, political, etc., design can sometimes be compromised. This framework is included in the cultural transformations brought about by the culture wars.

However, this secular context, this type of Western Judeo-Christian modernity, has not always safeguarded the conditions of this dialectical struggle and, at times, has protected itself against any transformation derived from cultural combat, with the aim of counteracting the experience of continuous provisionality generated in these scenarios. There have been episodes in which the diversity of moral positions and the struggle for centrality in social legitimization have been denied. In the case of Western modernity the cause has not (always) been sacral-religious fundamentalism. On the contrary, it could have been a "secular progressivism" (Gorski 2020, p. 64) articulated around the enormous strength of scientifically verifiable knowledge on the horizon of facts. It is important to point out that *this social discourse has colonized the narrative core of* Western *modernity and has defined its axiological scheme for a good part of it*. Its predominance has meant a certain reconfiguration of the universes of meaning in which scientific formalization was erected in *the* social language. In this sense, modernity has tended to be defined mostly *by opposition to the* dominant *transcendence* since the emergence of axial cultures (Jaspers 1994; Eisenstadt 1986) and their symbolic forms, such as religion, metaphysics, art, etc. This tendency has undergirded the institutional configuration of *the secular era* (Taylor 2007), giving rise to what Jürgen Habermas calls a process of *selective modernization* (Habermas 1998, vol. II) based on the power of the instrumental formalization of the world-of-life. However, this is neither a historical automatism nor a product of destiny. Modern expression and its secular spirit make a rigid and uniform social life impossible, even if certain radicalized tendencies pretend so. History is a horizon of unpredictable changes that announce the known and, at times, the unknown.

This paper analyzes the social processes that have led to the consolidation of a technocratic secular order and the type of cultural struggle that has made it possible in the Western Judeo-Christian programme of modernity. To this end, we first propose a reconstruction of the technocratic consciousness in the course of the secularization process, culminating in *technological determinism* (Morozov 2011, p. 295) or in the *technological solutionism* (Morozov 2013) of the social present; then, the analysis will dwell on the *neo-immanentist* (Gorski 2022) tendency of techno-functionalism, in which the secular context and the text of secularization become *one* and exhaust the social explanation; thirdly, this reflection deals with the open nature of secular life, in which context does not determine social texts (inter-actions) and opens the way for the existence of different *life options* (Joas 2012) that compete with each other and even rebel against institutional design. The reflection, then, focuses on the specific features of the culture wars in Western and Judeo-Christian culture and their globalizing tendency; finally, the paper closes with a conclusion in which the path taken and the new challenges that emerge from the arguments put forward will be reflected upon. It is not, therefore, a case-study analysis, but rather an investigation of one of the main axes on which current discourses and social interactions are articulated and which leaves its

imprint in various ways on concrete cases of culture war. This axis, neither exclusive nor excluding, is that of techno-functional neo-immanentism.

In short, our work analyzes how in secular societies a technological determinism (Morozov 2013) has emerged and crystallized, based on the articulation of what we have called a religious neo-immanentism (Gorski 2022) of a techno-functional character. That is, behind technological determinism we find not only numbers, data, or facts, but attributions oriented to logics of creation and articulation of meaning.

From a methodological point of view, this work is an in-depth critical–theoretical analysis of the existing literature in the social sciences on neo-immanentism and its techno-scientific drift in Western modernity. That is why, in order to carry out our work, we turn to secondary sources of quality and scientific impact—those mentioned in the previous paragraphs—to carry out, from these sources, an in-depth exploration of neo-immanentism's hegemonic character and its link to and impact on the culture wars.

## 2. Historical Reconstruction of the Techno-Functional Issue in the Secularizing Course of the World

In the development of sociology, the providentialist and teleological view of history has played a considerable role since the beginning. From this approach, modernity has been thought of as "a post religious temporal stage" (Casanova 2019, p. 14). This is a gaze that normalizes the presence of a scientific explanation of intra-mundane facts and the absence of religion in the institutional tissue of society from an almost natural course of secularization, inaugurated with the axial revolutions. To express this situation, José Casanova speaks of a "modern 'stadial consciousness', inherited from the Enlightenment, which understands this anthropocentric change in the conditions of belief as a process of maturation and growth, as a 'coming of age' and as progressive emancipation" (Casanova 2019, p. 15). The work of authors such as Taylor (2007) serves as a relevant expression of this tendency that returns recurrently in the course of Judeo-Christian modernity. The rationalized model of life is imposed, based on the autonomization of science, economy, and politics from the control of the church. This is a model alien to the expressive support of social life and centered on what Taylor defines as "buffered identities" (Taylor 2007, p. 27). In some sense, religion abandons the centrality of society and is relocated to the private sphere (Luckmann 1967; Beck 2010; James 1917).

In this scenario of scientific rationalization, the technical-functional domain of reality has emerged as one of the great axes on which (inter-)actions are articulated in modern social life. While in axial societies—from the emergence of second-degree thinking (Elkana 1986) and the establishment of a chasm between the transcendent and the immanent (Bellah 1969), which gave rise to what we know as universal (Weber 1978) or historical (Bellah 1969)—religions laid the basics for the later development of scientific societies, modern techno-scientific societies have unilaterally tended to identify themselves with the intra-worldly domain, thereby causing the erosion of the—until then—moral and cognitive superiority of the plane of transcendence. In turn, the same happens with *the languages of ultimacy* of the plane of transcendence such as myth, religion, philosophy, and art. The modern world disavows the guiding figures of transcendence that were characterized by a contemplative, attentive gaze, oriented towards answering the question of *what* the universe is. In the face of profiles such as priests, mystics, and philosophers, who embodied the beatific universe of the moral order, Western modernity promotes new social figures such as technicians, discoverers, inventors, programmers, etc., who ask themselves *how* the facts of the world work.

There may be three causes that act almost parallel to each other and explain the hegemonic crystallization of techno-scientific expansion in the course of Western modernity:

1.  In the first place, the emergence of Western and Judeo-Christian modernity takes place with a change of social values in which *the "affirmation of ordinary life"* (Taylor 2007, p. 144) gradually makes its way into prominence. What, until then, evoked the ephemeral character of earthly life and the prevailing degeneration in the intra-

mundane framework is transformed with the advent of modernity into the new moral support of society. The metaphysical depth of existence begins to lose the centrality it held in previous epochs and it is the plethoric richness of experience that claims a level of social attention that was hitherto non-existent. Material production and sexual reproduction are placed at the center of social attention. The gaze of bourgeois man begins to attend to intra-mundane processes over which the new society discovers an unknown capacity for (technical) control. At the same time, both the existence of a Judeo-Christian religion, in which history seems to autonomize itself from divine control, and the growth of scientific progress, expressed through a mathematical and semiological language (Reinhard 2006), converge in a horizon of action in which the most immediate challenges have to do with what a human being is capable of doing and discovering. It is the embryonic moment of the awakening of *human autonomy.*

2. Secondly, a mutation occurs within this secularizing logic in which *science* becomes the narrative of Western industrial modernity. Moreover, *the* narrative becomes one with the secular context. *Text and context* become confused. This is the era of *scientism*, in which other approaches to the world are marginalized. Questions concerning the meaning of existence, freedom, the limits of scientific intervention, death, and other universals cease to be scientific problems. In *Wittgenstein's Vienna*, as the title of Janik and Toulmin's book (Janik and Toulmin 1974) plastically captures, there is a clear analysis of how modernity conceives of these questions as *pseudo-problems* (in particular, for the early Wittgenstein). In this epoch, the world is described as a reality based on a recurrent order of facts that is representable by the abstract language of scientific thought. Modern Western society loses the spaces of moral reflection in which the actor debates about the emergence of the *one-dimensional man* spoken of by Herbert Marcuse (1991) and about the *conversion into myth of* technical rationality pointed out by Adorno and Horkheimer (2007). Undoubtedly, the *Dialectic of Enlightenment* is a core source of inspiration for subsequent analyses that study the technocratic drift of Western modernity. Reason turned into myth is one of the substrates on which the new techno-functional neo-immanentism feeds and, therefore, one of the fundamental elements when it comes to analyzing its impact on current societies. Calculation becomes a form of reasoning that does not require deliberation in the act of individual decision. The repetition of the sequence constitutes the core of the mental operation. Initiative and individual responsibility disappear. It is a time in which the incontestable hegemony of science pushes man "in search of meaning" (Frankl 2008; Adorno and Horkheimer 2007; Marcuse 1991) within an inexpressive and uniform horizon.[2]

3. Finally, at present, scientism has been consolidated and has given way to *technological solutionism* (Morozov 2013). With the recent irruption of new technologies, the Western and (late) modern human being has experienced a change of social horizon. Today he no longer only controls but *enjoys* the use of the technological artifact to border on the idea of perfection in everyday life. Control has given way to playfulness in a social context dominated by the *device of creativity* (Reckwitz 2017). The emergence of resources, such as the internet and all its applications and virtual platforms, allows for the establishment of a *creative and re-creative* relationship of the individual with the universe. The idea of *play and experience* takes root in a technology that not only solves problems in an instrumental way, but simply *entertains* (Debord 2011). A new, innocent consciousness emerges, freed from any hint of guilt and responsibility since it ignores the notion of boundaries. Taking for granted the control of the universe, contemporary human being sets out to enjoy himself in *The Metric Society* (Mau 2019), in which everything is translated to data "contributing to our gradual assimilation into the great social game of mutual evaluation, observation and comparison" (Mau 2019, p. 180). To Mau "the quantification of the social world is not just a particular way of describing society, but has an impact on three sociologically relevant [. . .] respects. Firstly, the language of numbers changes our everyday notions of value and social

status. The spread of the numerical medium is also driving forward the 'colonization of the lifeworld' (Habermas 1998) by instrumental concepts of predictability, measurability and efficiency. Secondly, the quantitative measurement of social phenomena fosters an expansion, if not a universalization, of competition, in that the availability of quantitative information reinforces the tendency towards social comparison, and hence towards rivalry.[3] [...]Thirdly, a trend is emerging towards further social hierarchization, in that representations such as tables, graphs, lists or scores ultimately transform qualitative differences into quantitative inequalities". In contrast to the apocalyptic rhetoric that accompanies the appearance of the computer in its beginnings, subsequently the discourse "eschatological because the computer is linked to problems of life and death" (Alexander 2000, p. 191) has been making inroads. In this context, transhumanism bursts into the philosophical debate with great timeliness. Although there are different versions of transhumanism, all of them share the idea that technology allows human beings to abandon passivity in the socio-evolutionary process and to take control over it (Dieguez 2021, p. 41). Dreams such as immortality, genetic selection, *biotechnological improvement* of the human species, and others, force different cultural sensibilities to take a position in public space. For example, positions such as those of communitarians and conservatives view these possibilities with distance, while technocrats and liberals imagine profitable future applications (Dieguez 2021).

### 3. Neo-Immanentism as the Return of the Sacred in the Secular Era

Despite the fact that, as we have just commented, in late-modern Western societies we can observe a clear rooting of the technocratic component, the historical past and its socio-cultural configurations refuse to disappear definitively (Bellah 2011). While Judeo-Christian modernity has enhanced the *statist vision* of the historical becoming (Casanova 2019) in which simple forms centered on the religious core inexorably give way to complex forms decentered and settled on the secular plane, current patterns of behavior call into question this gradualist and teleological approach to the evolutionary and cultural course of human societies. Social expressions that emerged in other moments of history that we thought had been overcome (Donald 1991) are opening up. In this sense, the sacred returns. Or, in other words, it does not disappear and, at the same time, it begins to acquire new faces. The objectivist dryness of technological society has also, in Max Weber's expression, its *enchanted garden.* Technology is erected in *hierophany:* in *technophilia.* Despite the decadent presence of the extra-mundane plane of transcendence, the symbolic density of the sacred illuminates intra-mundane spheres of Western social modernity, one of which is the technological artifact, whose analysis is the *leitmotif of* this work. As in the pre-Christian pagan period, the contemporary world is flattened, self-absorbed, and exhausted again on the plane of immanence. Specifically, Philip Gorski speaks of neo-immanentism in the current context in which "the world is imbued with the mystery and power of the sacred, and the concomitant decline of transcendent worldviews, which hold of the sacred to be wholly other and beyond" (Gorski 2022, p. 46). In a way, the new idolatry arises from the foundations of a consciousness of the Western modern human being that has become self-referential and has lost their relationship with alterity, which "only affects the aspect of freedom and not that of finitude" (Bell 1991, p. 32).

Although the current social models offer biases proper to our times, they also find elective affinities with cultural models that preceded the emergence of the axial religions described by Jaspers and others (Jaspers 1994; Eisenstadt 1986). In these, transcendence prevails as a legitimating instance of the established order.[4] In them, "the afterlife is portrayed as immeasurably superior to this life. Eternal salvation becomes a central aim of religious practice, for some *the* central aim" (Gorski 2022). If axial religions are characterized by a vision divided into two planes of reality, transcendence and immanence, in a dualistic way (Bellah 1969), pre-Christian pagan civilizations (such as Egypt, Mesopotamia, etc.) offered a worldview more clearly centered on the immanent plane of existence. Let us say

that in these (pagan) societies, transcendent forms developed in a context of immanence. In them, a *compact* vision of the cosmos prevailed (Voegelin 2001). Everything was *one.* God, nature, and society were integrated and fused in an undifferentiated continuity. In Weber's words, "the primitive cult of the community and, above all, that of the political communities, excluded all individual interests. The tribal god, the local god, the god of the city and the god of the empire, cared only for interests that concerned the totality: rain and sun, hunting, victory over enemies. Hence it was the totality as such that addressed him in the community cult" (Weber 1992, p. 238). In this *unitary background* governed *the power of the sacred* (Joas 2021) that flowed and moved throughout the cosmos. From it radiated an *effervescence,* also *sacred,* renewing the cosmic cycle and human life as part of it.

In these archaic social models (Bellah 1969), the idea of extra-mundane salvation does not apply. There is no more-there. The more-there exhausts what there is. In it appears the continuity of everything with everything, so that the modern boundaries between God, nature, and man are blurred. The ethical rigor of the axial religions oriented towards a model of consciousness that regulates itself on the basis of the moral codes of the universal does not exist in this period. Rather, a pattern of social authority charged with magic and numinous power predominates. It is the figure of the king endowed with a divine force. He stands out for his power of transformation in everything and over everything. For him, any prodigy is possible. In these divine monarchies the king is *like God*, "the power of the king is a portion of what is called in the island cultures of the western Pacific *mana*, the power that flows through and sustains life and the cosmos as a whole" (Gorski 2022, p. 50). In this process of incessant conversion, the king knows no limits. He acts *by enchanting and experimenting* with new forms of dealing with matter. He plays. Likewise, he is unaware of the notion of guilt and responsibility. There is nothing and no one above him to whom he is accountable. In him resides an overflowing potentiality that transgresses rigid social classifications and their fixed identities. He dwells outside logic and morality. He is a *strange king (god)* (Sahlins 2008) originating from an unknown world, alien to the normal schemes of behavior and, therefore, in a position to transgress and violate the instituted. Therefore, "Power, not righteousness, is the mark of his divinity. Nor does the king 'possess' his power within his person so much as channel it from the cosmos through his person" (Gorski 2022, p. 50).

Attuned to this cultural precedent, the current model of society could well be described as *neo-immanentist* (Gorski 2022). Let us see in what sense we can establish continuities (and discontinuities) between these two periods. If in the archaic-pagan period sacred authority shows a protean dimension that made it indefinable and strange, today this dimension is enhanced in light of the unlimited technological and virtual potentialities of our time. In a way, technology becomes king, or becomes its technical substitute. The protean drive, previously only in the hands of the king, is democratized. The magical thinking of those enchanted cosmovisions of the old divine monarchies is once again making its presence felt. This is clearly perceived through the fact that technology does not impose itself because of its scientific character, but because it has become a way of enchanting (or re-enchanting) the world. The imaginary of technique has become a myth in our societies. Empirical media is no longer a guarantor of scientific veracity. Now, individuals and institutions incorporate it into their lives through the magnetism of digital indicators. The frontiers between reality and fiction are dissolving. The contemporary human being is not afraid of technology, but rather inserts it into his daily and corporal reality. He adopts it as *another sense* that allows him to see, imagine, speak, and decide remotely. The resistance of matter and physics are overwhelmed by the *numerocracy of indicators and parameters* that "make us more predictable, but also more calculating" (Heintz 2010, p. 176). Dataism and mathematical grammar, already settled in the modernizing course of the previous domains of instrumental reason, are erected in the narrative, but "they are not copies of a pre-existing reality, but selective constructions partially responsible in the creation of that reality. The objectivity of numbers is not so much a fact, as an attribution" (Heintz 2010, p. 170).

The possible implementation of neo-immanentism in our time is characterized by the following features:

1. In a world tending towards the secular, *the sacred returns*. It does so, just as it did in pre-Christian (and pre-axial) societies, in the form of a blind, anonymous, and magical force that bursts forth from a domain inaccessible to human cognition. It is the mystery incorporated in the intra-mundane plane that sustains the social order and, by the same token, transgresses it by overflowing its boundaries. Its pure non-being, nothing in concrete, is the condition of possibility of being able to be anything. The sacred is presented as the secret prodigy of technology.

2. The force of the sacred arises from *the indistinct and the undifferentiated*. In it, the clear distinction of logical and ethical concepts is blurred. Its use is not so much the product of thoughtful deliberation as of *improvisation* and *occurrence*. It inhabits a domain devoid of distinctions and, at the same time, that generates distinctions. In it, good and evil coexist in a never-ending war (a conflict, a tension). Works incorporate developments in which plenitude is followed by tragedy.

3. Digital technology becomes *symbolic hierophany* (Eliade 1981). In it the actors prolong their dreams of control and (playful) experimentation, since nothing is impossible to discover and realize. Human potential is now unknown, as is the mystery that pervades social reality. It resembles the absolute power of those divine kings of the old pagan religions, and the power that enabled them to exceed all that was established. The human being has been given the opportunity to experiment bodily and emotionally with a technological instrument saturated with magical power.

4. In this context, there is a proliferation of social profiles in politics, culture, public opinion, etc., that attract social attention because they represent the *absolute power to transform reality*. It only seems to be in their hands. In them, the mystery of the plenipotentiary force of the sacred and its enormous capacity to respond to the uncertainties of the moment becomes visible. Their virtue consists in simplifying the enormous complexity of global societies from a confusing and turbid mixture of non-contrasted assumptions, half-truths, information saturation, and discredit of truth and thought. It is the time of embezzlers and tricksters in politics and public opinion.

5. "Neo-immanentism is a global phenomenon" (Gorski 2022, p. 49). It is constitutive of a model of life that exceeds cultural limits. Although its origin is in the Judeo-Christian West, which we analyze in the present work, today it is part of the symbolic capital of humanity. The lights and shadows derived from its global implementation are shared on a planetary scale. The role of remote communication and the emergence of new forms of authoritarianism are its most visible features.

In this sense, technology is charged with *aura* (Benjamin 2021). In it mature the great hopes of the contemporary human being to master, but, above all, to *playfully experience* its digital resources. The phase of technical mastery of the world was already overcome at that moment of intense debates about the limits of the world between the representatives of logical Positivism—defenders of the theory of the linguistic representation of experience—and those of the Frankfurt School and Existentialism, more akin to the social-historical mediation of human action in the definition of things. *Now it is a matter of having technical mastery for testing, discovering, and entertaining*. A hybrid version arises from the tension between the controlling pulsion and the playful one. *Its dataistic language translates the objective facts, but also the subjective processes of the actor. Here is the novelty of our time*. Dataist comparativism defines the contours of our horizon of social action. Its virtual platforms include ratings and rankings in the context of sports competition, academia, the economy, etc., but also in the context of our preferences and desires. We live with big data and personalized marketing. We know about ourselves through the one who knows the most about us: the virtual network. Data reveal unknown dimensions of our biography. They make us see the invisible, know the unknowable of our intimacy and of the comparative relationship with others in different global contexts of action. What is important in this case

is the numerical pattern of this expressive background, which is silenced. It has become an objective parameter that orients behavior in a global horizon saturated with information.

In this process of technological orientation of action, algorithms[5] are of utmost importance. Based on the frequency and direction of our virtual searches and tracking, they provide the secret constants that express our preferences. They reveal the common denominator of their daily use. They become, in this way, an illuminating resource and a criterion for individual decision. They are used in all instances of social action and decision as referents of symbolic objectivity. The key to understanding their impact on the background of today's societies lies in their capacity to offer meaning and a logic of action. The algorithm has a *revealing* dimension: *it announces the future that awaits us.* In a way, it is the warm home that brings us certainty in a hypercomplex and bewildering horizon for decision making. Science in different areas such as health, the sustainability of the planet, care, bodily routines, etc., anticipates the immediate future of our lives. This points to the determination of *its future flow.* It would be a sort of immanent divinity (fatality) that anticipates the course of events, freeing us from having to decide in this global jungle of information. In a way, it only demands fidelity to its announcements in exchange for freedom from decision. Therefore, to the revelatory dimension we must add another: the soteriological one.

Helga Nowotny's (2022) studies on the role of artificial intelligence (AI) in our way of life put us on alert about this danger. Along with its many possibilities in terms of solving technical difficulties, the problem consists in granting *the last and only word* in these times of *technological epiphany* to the algorithm, to the detriment of human reflection. AI knows a lot, but not everything. In particular, it does not know the components that shape it and the social context in which its procedure is applied. In Nowotny's words: "algorithms will have to acquire culture" (Nowotny 2022, p. 131). They always operate following a predetermined programming, with no capacity to respond to possible setbacks or contingencies given in the context of application. It lacks, therefore, self-reflexivity. Thus, the sacred hegemony of technology can be translated into a new version of deterministic dominance over human decision. This is what Morozov (2011) refers to when speaking of technological determinism.

In fact, voices are beginning to emerge from civil society calling attention to the unlimited power acquired by AI. As published in the Spanish newspaper *La Vanguardia*, the first citizen lobby has just been formed with the aim of defending human interests in the face of AI. According to the information, "it is called Civicai (civic + ai, civic artificial intelligence) and seeks to organize ordinary citizens to take a stand and make their voice heard collectively in the face of the changes implied by the use of this type of technology[6]" (Rius 2023, p. 24). Once again, the culture wars make visible the struggle for the symbolic centrality of society. The legitimacy of the neo-immanentist digital framework is called into question. The number of narratives in the face of this reality is growing. Its dangers in the form of determinism enter fully into the public debate. Likewise, ChatGPT is generating an immediate reaction within the teaching community of schools and institutes due to its unlimited capacity to generate (apparent) evidence through artificial intelligence alone. Its competition makes it difficult to identify the authorship of texts. This is a new scenario, in which progressive and conservative forces seek to have their voice recognized in the civil sphere, as Hunter (1991) points out. This implies that around the use of AI applied to education—ChatGPT—a cultural conflict is also being articulated. This is a conflict that is established mainly between the advocates of the benefits of technological development and the need for openness to such 'progress', and those whose narrative focuses on the 'dangers' associated with such developments.

Nowadays everything is data. Moreover, everything is digitally recorded as data. The facts of the world are agreed in an objective and neutral format. Both objective and subjective facts share the *cult of evaluation* "making visible what is invisible" (Mau 2019, p. 99). The polychromatic entrails of experience are represented by numbers and computations. The objectivist translation of empirical abundance suggests a total mastery

of the course of events. The algorithm makes the future predictable. Society surrenders to it as a pattern of revelation, even, at times, as a prophecy that points directly to possible courses of action. In a way, it thinks us and defines the way we represent reality. The margins of human decision are contracted. The individual is liberated, although he also submits to a new coercion of an inexorable of neo-immanent origin.

The background of this scenario is defined by the dominance of an extreme version of victorious secularization. In this social context, the strength of secularization is self-affirmed with the help of a *secularist* narrative (Casanova 2019), that is, of a narrative that enhances and mythologizes the liberating scope of scientific knowledge as the *only* explanatory model of society. Modernity is based on the "self-referential character of immanence" (Verhoef 2016, p. 4). Context and text become *one.* They con-fuse. Social debate is impoverished to near extinction. The Other of the technological representation of the world is absent. Plurality suffers. Other symbolic positions, such as the arts, metaphysics, religion, mysticism, and other social profiles such as priests, philosophers, artists, and laymen, are left out of the frame of representation. Their existential contents cannot be translated into data. In the face of this, they reveal themselves and propose an approach to experience from the ultimate values. They express themselves through the *symbol*, in which no vision is definitive and ultimate. The reason–faith antagonism is at the basis of this framework. The first term of dualism has uncontested dominance because in technological reason resides modern man's dream of an exhaustive control of the cosmos. In the neo-immanentist picture there is no beyond. There is only an intra-mundane plane in which faith in the ultimate cognoscibility of experience by virtue of the revelatory potential of the secular deity of the algorithm rules.

The universe is contracting under a form that borders on the limits of a *digital fundamentalism.* Not for nothing does academia already speak of *technological totalitarianism* (Schirrmacher 2018). In it, experience is impoverished by its strong dependence on the empire of indicators. The data have constructed a "comparative panopticon" (Mau 2019, p. 159), in which a narrative is globalized within the reach of individuals based on numerical comparison. The auguries of Adorno and Horkheimer seem to be confirmed. In this way, doubt, search, imagination, and listening are absent. There is no more. There is no otherness because there is no *measure or* indicator for it. However, data cannot do everything. It resists the interpretation that social life obliges it to provide. Without the other, without otherness, there is no debate, only uncertain self-confirmation that fears the opening towards the conditions that make the existing context possible. Beyond the self-referential drive of technocratic neo-immanentism, contemporary challenges demand other voices and accents. The globalized world, and its wars, its pandemics, and its economic crises, reminds us of the need for *subtle languages* (Taylor 2007) in which imagination announces possible horizons of coexistence. Religion, like technology, only threatens if it excludes. Furthermore, contemporary social life anticipates fertile struggles between the two.

## 4. The Tension of Narratives of the Secular Context

As Hunter (1991) says, secularization occupies the progressive position of Western modern life in the culture wars. It does so repeatedly, with arrogance, by pushing alterity and its questions of ultimacy to the margins of the debate. Religion and positions of transcendence have become lost echoes in cultural debates. Somehow, as Gorski (2021) says, religion has been invaded by the political sphere, questioning the modern logic of separation of spheres and turning faith into an instrument at the service of the will to power of short-termism and reductionist discourses. The secular era has hypostatized the progressive position to the point of becoming not only the hegemonic mode, but, at times, the only social narrative. There is neither opposition nor space to question scientists and technologists about their decisions, to carry out exercises of second-degree reflexivity about the tasks they perform and, also, about the consequences that such exercises provoke in social life. We technicians (often including social researchers) seem to have lost sight of

one of the maxims of qualitative social research: the need for the researcher to question the biases he or she introduces into the analysis. We often forget that the researcher shares nature—sociologically speaking—with the phenomenon under study. Moreover, this gives the impression that only articulating this possibility would mean profaning the space of the sacred, forgetting that it is not only advisable but necessary to rationally justify the findings and technological applications among specialists and between specialists and society. In this sense, *technology represents social legitimacy*. Just because it is technology, its decisions should be accepted without question. However, if it is purely monopolizing the whole of social reality and restricting access to the truth of other cultural positions, the technological dream becomes the center of decisions, and their goal is to persist. Somehow it discovers in itself a conservative pulsion that denies its commitment to criticism and social nonconformism. *In the* neo-immanentist *context, digital 'fasto' identifies virtual network and reality*. Religion and other transcendent symbolisms are marginalized. Or, rather, their symbolic and indirect approach is undervalued. The linguistic faculties that reside in human representation are thus split to the point of incommunicability. Hyperformalized grammar has distorted philosophy, religion, and public opinion as universes with greater expressiveness and semantic density. Speculative abstraction has become recused from the qualitative experiences that situate human beings in the world (Jung 2014, p. 186).

However, new reflections on the (supposed) teleological course of the dominant secularization are taking place. In them, the unilateral background of the digital pulsion is not so much assumed. The problem is more complex. The social theorization of one of the most representative voices on the question of the religious fact, Casanova (2019), reveals aspects related to other developments of secularization that confront the digital hegemony of our time. In his recent research on the religious beliefs of our time, he states that secularization can lead to a (neo-immanentist) context lacking transcendent references, but it can also lead to a pluralization of beliefs and social transcendences. In the case at hand, neo-immanentism of a techno-functional character would be a belief that is not only clearly neo-immanentist, but one that tends not to recognize in terms of equality the plurality of modern and secular forms of transcendent being. A secular background does not recognize rigidity and determinism, although there may be positions (religious or secular) that advocate it. In this sense, the first model refers more to the European tradition; the second to the American one. While the former has dominated since the beginning of modernity, the latter is beginning to play a leading role in this globalized society. In it, the *immanent frame* begins to accommodate different religious sensibilities (axial, pre-axial, and post-axial) and secular sensibilities (atheists, scientists, technocrats, etc.) that coexist and share the same *community of destiny* for the whole of humanity. What Casanova defines as *global denominationalism* refers to this new situation, in which the secular and the religious generate codes of reciprocal recognition and in which they do not go separately, but define *in unison* the new time in which we live. In his view, "Global humanity is becoming simultaneously more religious and more secular, but in significantly different ways, in different types of secular regimes, in different religious traditions and in different civilizations." (Casanova 2019, p. 65)

This reflection allows us to approach a fact that contrasts with the initially statist consciousness of Western modernity, which is at the base of the dominant technological 'aura'. *While in modernity text and context become one, the secular context of our time is open to a relationship of conflict and confrontation between the ultimate positions of social groups without excluding anyone from the search for the symbolic center of social legitimization*. As Josetxo Beriain points out, modernity can be defined as a "set of provisional notes" (Beriain 2005, p. 6). Different *vital options* (Joas 2012) that feel challenged by the questions of our time arise and compete (Joas 2012). Today, more than ever, cultural struggles are open. Actors are making choices. The world of technologies does not answer everything. Cultural sensibilities are looking in other directions. The question of legitimacy remains open and without a definitive end. The sustainability of the planet, the multiple positions of the feminist struggle, animal protection, global fiscal balances, universal justice, etc., become referents

that mobilize communities in their cultural struggle for recognition and for obtaining a voice in the civil sphere (Alexander 2008), a public square where different socio-cultural sensibilities are debated and even confronted. In this vision of the secular, the symbolic center is not the property of anyone, it is not a destiny assigned by a deity or by a historical fatalism. Rather, it is the result of a contest in which social groups struggle in a scenario of freedoms that guarantees the right to consensus and dissent. In this scenario, positions are diversified on moral bases that are unquestionable for their members. *Global modernity is both secular and religious at the same time.* This step, in terms of inclusion, breaks the rigidities of the first phase of modernity. Or, at the very least, they are visible for everyone. The secular context is no longer mechanically self-affirming in a dominant narrative. There is conflict and struggle, as Weber rightly proclaims throughout his work, and there are worldviews that approach social issues from very different cultural perspectives. Humans speak, not gods, although in their narratives the initial impulse comes from scenarios charged with transcendence, which obliges us to guide and filter particular positions on the public stage. Today, more than ever, the *immanent frame* is the stage that accommodates different symbolic positions that enter into a tense relationship. If something persists at the heart of modernity, as Weber well knew, it is tragedy.

The redesign of the secular age just described introduces changes of enormous relevance in the image of society and in the meaning that emanates from it. Its religious–secular nature implies the existence of a plurality of accents that share the same horizon of coexistence. Moreover, it shares more: the *imaginary* origin of all cultural positioning (Castoriadis 2013). At times, the overflowing passions of the conflicting sides of the culture wars strain the social relationship to the extreme. Science and technology have always appeared in the eyes of public opinion as dispassionate and neutral worlds that omit the passionate impulse, but what we argue in this paper is that, converted into neo-immanentism, they cross these alleged barriers and access the space of meaning and value in which Weber placed the main challenges of the social sciences. This presence would only be part of the religious sphere. The political struggle is dialectic, but dialectic is always driven and directed by an emotional stimulus. If, in these culture wars, positions overflow and relationships are strained, the positive effect of these wars is the public revelation of extraordinary forces that unleash the social action of each group[7]. Science also rests on its *enchanted garden*(er). An idea of perfection runs through the technological imagination. Ideas of *limitlessness and plenitude* are not the exclusive patrimony of the religious world. They are also of the secular world. Hunter (1991) himself says that both progressive and conservative positions involved in the culture wars have *a religious origin* because they are based on *ultimacy. From there*, coexistence is transformed into a war between gods (Weber 1988) that ignores any glimpse of reconciliation.

Today, the technological dream imagines situations that until recently seemed simply unthinkable. Transhumanism, in its most radical version, defends the possibility of transferring virtual life and all its advantages to mental life, that is, turning human beings into a digital intelligence connected to the universal network and all its resources and virtual platforms. The cognitive lucidity activated by the digital network would coordinate mental and bodily processes. The body itself would become a dispensable reality. Only the virtually reconnected soul would be enough to define the *new human being*. This is similar to medieval thought, wherein the body was considered the *prison of the soul*. It is true that in human experience everything begins as a dream and then becomes reality. However, as Bell said earlier, we should not forget finitude as a structural feature of every human being. Only in this way would the inflamed technological world be deconstructed and, with it, its passions and dreams unveiled. Neo-immanentism expresses a sublimated universe in which the human being's technological resource is a symbol of sanctity. As Morozov says, ""The Internet", thus, is believed to possess an inherent nature, a logic, a teleology, and that nature is rapidly unfolding in front of us. We can just stand back and watch; "the Internet" will take care of itself and us. If your privacy disappears in the process, this is simply what the Internet gods wanted all along." (Morozov 2013, p. 24).

In a way, paraphrasing Karl Marx, we could speak of *Technological Fetishism* (Morozov 2011, p. 311): "everything that "the Internet" touches automatically gets better, smarter, prettier" (Morozov 2013, p. 26). The Internet has taken on a life of its own. It rules with autonomous power. Today it has become a kind of magical power that transforms whatever it touches. Its existence is seen in the eyes of society as a new eternity that does not belong to this world, that has never been born, and that is not to perish. It is the new promise of earthly salvation that explains everything because in its eyes everything is transparent or becomes so. In short, *the neo-immanent world is self-enclosed in a horizon directed by technology*.

However, the secular western context includes other religious and secular voices and accents that force it to qualify the affirmations of technology in order to adapt them to plurality. The latter must provide accountability and pedagogy within a society with a higher level of education than in previous periods. These questions are still present in public opinion: how far can/should technology intervene in the genetic components of a human being? Does the human being have moral limits that should not be exceeded? How far can the will of certain sectors of society go to turn technology into a genetic selection market at whim? The debate scenario presented by Ridley Scott in *Blade Runner* is taking shape in today's societies.

The sacralization of technology in the neo-immanent context reduces reality to data, to various formulas governed by an algorithmic pattern. The doubt is whether this instrument can answer the kinds of questions and concerns that arise in every human being. In a way, an old vice is reproduced. Religion lived through many centuries of domination in which it was the only social narrative. Worldly questions related to the control of reality were not part of reality. Today the relationship is reversed. However, in a secular context open to pluralization all sensibilities can be included and can intervene in the culture wars. A human being does not only need *truth* based on the adequacy between words and things. In the same way, *meaning*—understood as an element that integrates the fragmented experience of western modern life into an overall vision—is also part of existential questions. Truth claims in an objective relation to facts, whereas meaning refers to a subjective identification with experience. This is something that Weber already knew, and which differentiates the social fact with respect to other phenomena of nature. Hence, he created his proposal for a comprehensive sociology articulated on the basis of the meaning of actions (Weber 1978). Both dispositions are part of social life. The error of our time consists in trying to suppress the tension between the religious and the secular in favor of the latter. However, technology and religion, science and faith, seek their space in our time. They do not need a context in which they understand each other. It is enough that they do not neutralize each other, that they listen to each other and can assert their convincing capacity. *The symbolic center no longer belongs to anyone, yet everyone strives to approach it.* This is undoubtedly one of the paradoxes of the societies in which we live.

## 5. The Cultural Struggle in Western Judeo-Christian Modernity

Once the mark of secularization has been detected, particularly in a context of coexistence without a sacred center and openness to the permanent opening of cultural tension, reflection enters into another scenario. It is no longer a matter of excluding one part of the secular/religious binomial, but of analyzing this binomial in light of the social context itself, in which legitimacy is at stake. *The text of our time is not only secular and technophile.* There are others that attend to the infinity and depth inherent to the limits of the world. The secularized context does not necessarily have to fall back on secularism, that is, on a sort of hegemonic value focused on the logic of increasing technology and its liberating potential (Casanova 2019). Quite the contrary, it offers enough elasticity to include positions with very marked and antagonistic moral foundations. The secular and religious narratives of our western context have different versions. In the former, there will be room for atheists, scientists, skeptics, technophiles, etc., in the latter, believers, the faithful, the orthodox, etc. However, it is not only these positions that are in dispute in western modern societies. The context invites a great proliferation of sensitivities that are articulated in the form of

multiple debates and diverse ways of understanding and approaching reality (feminists, ecologists, pacifists, etc.) depending on the theme or problem to be addressed.

Techno-functional neo-immanentism has gained strength in the contemporary imaginary, and one of its objectives has been to agglutinate the meaning of social action around it. It has not succeeded, among other things, because there are different socio-cultural positions that articulate meaning around other spheres of social action, such as the sublimation of the spiritual, the divine, or the nation, to give a few examples. There are still vital options that base their images on some form(s) of transcendence. In other words, these are cultural outlooks that project the sacred beyond "the limits of mere reason" (in the Kantian expression). *The secularization of the context does not necessarily lead to the secularization of the text*. It leaves room for diverse sensibilities that are recognized in the secular/religious conflict (Hunter 2009). Issues such as abortion, euthanasia, women's rights, immigration, etc., often integrate the centrality of this binomial (secular/religious) into their struggles, since this is perceived as one of the axes on which the disputed positions in the civil sphere are articulated.

For all these reasons, it can be concluded that, within the framework of a growing trend towards globalization, the axial civilization of the Judeo-Christian tradition emphasizes the secular/religious binomial as the axis of its mode of cultural struggle. It is its particular contribution to a field of struggles that enters and expands on the stage of globalization. At the same time, as Gorski points out, neo-immanentism is a phenomenon that expands globally. Just as we can find other binomials in different cultural contexts (in China, for example, the techno-nationalism of the Communist Party versus the expressions of the capitalist West; in Russia, the secular imperial Great Russia versus the bourgeois West; in Iran, the Islamic-based political religion versus a pro-Western global youth), in the Judeo-Christian West—the context we have analyzed—the secular/religious opposition intervenes in a high proportion of the cultural debates. It is not the only one, but it is true that its influence has a great impact on the rest of the binary classifications. The symbolic antagonism between the binomial religion-conservative/secular-progressive (Hunter 1991) makes visible the cultural position of groups in any debate. This is the basic distinction in the cultural wars of the West. Although it is added to other binomials, it is the fundamental axis that allows us to be seen and to understand the movements of the rest of society. It constitutes the way of looking at our reality and that of the rest of the planet.

This binomial is not a recent development. Religion has always been part of the formative processes of Western Judeo-Christian modernity. Its first state forms and their corresponding processes of confessionalization, the emergence of the (Protestant) identity alone with God, the providentialism of Enlightenment history akin to the religious providentialism of the Christian religion, among others, explain the profound connection between a secular sphere that seeks its autonomy and a religious sphere that forms part of its basic core. It is true that the peculiar bias of Western history has possibly been the emphasis on *the functional differentiation* of value spheres and, in part, differentiation between the secular and the religious. One of the clearest manifestations of this drive for differentiation is found in the dual clause of the First Amendment to the American Constitution which embodies the principles of "no establishment of a State religion" and "free exercise of any religion." However, social developments continue to occur in which the secular and the religious, more particularly the state and the church, once again come dangerously close (and de-differentiate). An example of this (although not the only one) takes place when the ecclesiastical institution legitimizes and protects authoritarian political forms as is the case of the figure of Patriarch Kirill in currently justifying Russia's military aggression against Ukraine (Casanova 2023, p. 81).

This process of neo-immanentist predominance is chronicled in sociological theorizing itself under the binomial *tradition/modernity* (or also *extra-mundane transcendence/intramundane change*). (Social) science continues to fall prey to its influence. The current deconstructivist tendency tends to reduce transcendence and its different symbolic expressions to a sublimation of the immanent. All extra-mundane idealization is explained as

a compensatory sublimation *of the only thing there is: empiria.* In this way, the explicitly unacknowledged assumption of deconstruction, immanence, explains everything else, in particular, transcendence. In the genealogical line of Nietzsche and, later, of Foucault, the deconstructive task seems to end when the *historical condition* of any idea of eternity (and tradition) is unveiled. However, as Joas (2011) says, the revelation of the conditions of emergence of ideals does not reduce their catalytic dimension of value and stimulus to collective action. This is not because they have been deconstructed; they cease to have value as a symbolic construction in the cultural world. They are operative and mobilizing. The fact of scientifically explaining a social process from *the social* (Durkheim 1982), does not imply that it is invalidated. Value beliefs can be explained and continue to be operative in cultural life. In this sense, the analysis of deconstruction as a methodological guideline of neo-immanentism is still pending. It is not in vain that deconstruction also has its *immanent*, *contextualized*, *and imaginary conditioning*, that is, its *mythical* seat that is hidden under the dominant narrative of the superiority of a way of living akin to the technological (and technocratic, this is, techno-functional) domination of the world. Deconstruction could also be an *imaginary kind of construction* in need of some form of creativity and transcendence.

## 6. Conclusions

In this secular context of strong neo-immanentist imprint, cultural expressions are not exhausted by technological attractiveness. It enjoys a dominant position, yes. In it coexist the desire for control, of which the Francfortians spoke to us, and the desire for entertainment that Reckwitz (2017), among others, emphasizes today, without forgetting Debord (2011). However, at this crossroads of technical and aesthetic pulsions, civilization produces paradoxes and contradictions that, to this day, have not found an answer. The distance required for the neutral study of reality by technology interacts with a torrential emotivism that demands from the world experiences of self-referential singularity. Between the question of *how* and the demand for *emotional visibility there* is nothing. If anything, there is a vacuum that is filled by profiles of a markedly *authoritarian and radical* character that emerge in a cultural conflict devoid of agglutinative and deliberative positions, which understand freedom *against* society and not *from* society (Amlinger and Nachtwey 2022). The gear formed by a technological activity, mobilized by emotions on the surface, only needs actors defined by their capacity to react, without the presence of other human capacities that allow society to see beyond the acts and decisions.

It would be no small achievement, however, if neo-immanentism, in addition to grappling with other cultural sensibilities, would be able to be in dialogue with its own—supposedly dispassionate—moral foundations. These are as imaginary as those of its adversaries. This step would be the best guarantee that the cultural conflict continues, but not always from self-referential, reactive, and conclusive positions.

Although the relationship between techno-functional neo-immanentism and emotivism is an avenue to be explored in future research, in this conclusion we would like to briefly point out that the work carried out has tried to reveal the religious-transcendent dimension that is hidden behind aspects of social life that are presented as lacking this dimension. Throughout the text we have become aware that the imperative of data or the invasion of everyday life by the algorithm have made these—data and algorithm—capable of transcending—going beyond—their factual logic, to access the realm of attribution and meaning. This issue has an impact on the balance of forces in different scenarios of the culture wars.

**Author Contributions:** Conceptualization, C.S.C., J.G.-G. and P.E.E.; methodology, C.S.C., J.G.-G. and P.E.E.; software, C.S.C., J.G.-G. and P.E.E.; validation, C.S.C., J.G.-G. and P.E.E.; formal analysis, C.S.C., J.G.-G. and P.E.E.; investigation, C.S.C., J.G.-G. and P.E.E.; resources, C.S.C., J.G.-G. and P.E.E.; data curation, C.S.C., J.G.-G. and P.E.E.; writing—original draft preparation, C.S.C., J.G.-G. and P.E.E.; writing—review and editing, C.S.C., J.G.-G. and P.E.E.; visualization, C.S.C., J.G.-G. and P.E.E.; supervision, C.S.C., J.G.-G. and P.E.E.; project administration, C.S.C., J.G.-G. and P.E.E.; funding

acquisition, C.S.C., J.G.-G. and P.E.E. All authors have read and agreed to the published version of the manuscript.

**Funding:** This research received no external funding.

**Institutional Review Board Statement:** Not applicable.

**Informed Consent Statement:** Not applicable.

**Data Availability Statement:** Not applicable.

**Conflicts of Interest:** The authors declare no conflict of interest.

## Notes

[1] The work we present focuses on analyzing neo-immanentism in the context of Western societies of Judeo-Christian tradition. It would be interesting to analyze other cultural programs in which modernity has materialized and to establish a comparative logic of differences and similarities between them. We consider this a task to be developed in future writings.

[2] The thesis we defend could also find support in the work of philosophers like Ernst Cassirer, who analyses the symbolic way of being of the human being (and, therefore, also of all his cultural creations: myths, religions, arts, languages, sciences, and techniques) or the hermeneutics of Heidegger and Gadamer, according to which interpretation is constitutive of the way of being (hermeneutic) of the human being that is distended between the realm of truth and that of meaning. Both philosophies come to recall, pointing in the same direction as this paper, that consciousness (in this case techno-functionalist consciousness) does not sustain or generate itself but arises within processes that are not strictly conscious. In short, Martin Heidegger's work on technique in the diagnoses of modern life, for example, occupies a prominent place in contemporary philosophy. However, to delve into these issues would divert us from the focus of our work. A contribution on neo-immanentism from a Heideggerian perspective remains to be made.

[3] This clearly has a direct impact on the culture wars phenomenon.

[4] Despite the symbolic predominance of transcendence over immanence in axial civilizations, transcendence is not perceived in the same way in all of them. While in Judaism, Buddhism, and Hinduism it is situated outside the limits of the world, in Confucianism and Taoism it is situated on the intra-mundane plane. In the words of Francois Jullien, transcendence in Chinese culture involves "the totalization of immanence" (Jullien 2019, p. 91), the holistic vision of the antagonistic forces (Yin/Yang) that energize the course of the world.

[5] To exemplify this empire of the algorithm or data, we would like to give an example: for some time now, sports teams have been basing their strategies for signing and selling players on protocols in which what dominates is a mathematical or algorithmic logic of action based on numerical scales that has banished that of professional advice that was articulated through the experience of a subject matter expert. It goes without saying that such algorithmic-mathematical logic is less subject to error than that which was articulated through the expert (who used to be a former professional in the sport in question).

[6] https://www.lavanguardia.com/vida/20230321/8838266/nace-primer-lobby-ciudadano-defender-intereses-humanos-despliegue-ia.html (accessed on 21 March 2023).

[7] For more information about this idea see Boltanski and Thévenot (2006).

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
