# Peer review of "The Hegemonic Character of Techno-Functional Neo-Immanentism and Its Relationship with Culture Wars"

_religions, doi:10.3390/rel14070943_

Round 1

Reviewer 1 Report

I. Clarity and Relevance:
The study's title and abstract offer a sophisticated exploration of techno-functional neo-immanentism and its connections with contemporary culture wars. The research delves into critical aspects of technocratic consciousness, secularization, and the cultural implications of these processes. It's a timely and relevant topic given the increasing prevalence of technocratic thinking and its impact on culture.

II. Methodology:
The paper seems to be a critical analysis of existing literature and theories rather than an empirical study. While the abstract and conclusion imply an in-depth theoretical and philosophical exploration, a clearer outline of the paper's methodology is needed. It would be beneficial to detail how sources were selected, how arguments were developed, and the theoretical underpinnings of the analysis.

III. Limitations:

Language: While the paper is erudite, the language can be dense and complex. This might limit its accessibility to a broader readership. Simplifying the language without losing the depth of analysis would increase its reach and impact.

Lack of Empirical Evidence: While the paper offers a theoretical exploration, it could be strengthened with empirical evidence to substantiate the arguments made. This could be in the form of case studies or data analysis that exemplify the trends and processes discussed.

Limited Contextualization: It is unclear whether the paper considers the global diversity of secularization processes and culture wars. As it stands, the focus seems to be on Western Judeo-Christian culture. Expanding the scope to include other cultural or geographic contexts would enhance the study's comprehensiveness and relevance.

IV. Recommendations:

Clarify the methodology: Provide a detailed account of how the analysis was conducted, including the selection of sources, the development of arguments, and the theoretical grounding of the study.

Simplify the language: While maintaining the depth of analysis, strive to make the language more accessible to a broader audience.

Incorporate empirical evidence: Substantiate the theoretical arguments with empirical examples or data analysis. This could take the form of case studies that illustrate the processes and trends discussed.

Expand the context: Consider incorporating a wider variety of cultural or geographical contexts in the analysis of secularization processes and culture wars.

In conclusion, this study offers a thought-provoking exploration of techno-functional neo-immanentism and its connections with culture wars. With some revisions and additional clarification, it promises to make a significant contribution to the field.

Language: While the paper is erudite, the language can be dense and complex. This might limit its accessibility to a broader readership. Simplifying the language without losing the depth of analysis would increase its reach and impact.

Simplify the language: While maintaining the depth of analysis, strive to make the language more accessible to a broader audience.

Author Response

Dear, reviewer and editor

We would like to thank the reviewer for his thorough reading and recommendations to improve the text. We would like to point out that the additions or corrections related to reviewer 1 are those that we have highlighted in the text in blue.

Improvements according to the reviewer's suggestions:

  1. Clarify the methodology. We have introduced a paragraph expressing the methodology used and noting that the main sources used to carry out the analysis are those referred to in the final paragraph of introduction.
  2.  Simplify the language: We have reviewed the entire text, removing expressions that could be excessively erudite. We have not highlighted it in blue because it affected practically the entire text, and it could be confusing to identify the changes.
  3. Incorporate empirical evidence: Although we consider that the text already presented an example -the formation of the Civic-ai collective in Spain in the face of the excessive presence of AI in our daily life- of cultural tensions between progressive and conservative views with the background of techno-functional neo-immanentism, we consider that, in accordance with the reviewer's consideration, it would be interesting to add another example. In this case, we have added on page 11, lines 413-422 an example linked to the emergence of ChatGPT in academic life and the narratives that have emerged in favor and against its presence in institutions such as the university.
  4. Expand the context. With respect to the advice to expand the contextual scope of the paper, we consider that the conclusions we have presented make sense in the scenario analyzed, therefore throughout the text we have reinforced the idea that what has been commented can be applied to the scenario of the Western and Judeo-Christian cultural program, outside of it, most of the conclusions may not have sufficient consistency. However, in order to satisfy the interesting reviewer's demands, we have introduced a footnote (number 1) to point out that this paper is focused on analyzing the Western Judeo-Christian program, but that it would be interesting in future research to analyze the similarities and differences between this cultural program of modernity and others, because we can say that techno-functional phenomena is global. The doubt appears in the case of the impact of neo-immanentism in other cultural program.

If the reviewer needs further clarification, we will be happy for answering.

Kind regards,

Reviewer 2 Report

This paper raises the highly topical problem of the relationship

between techno-functional neo-immanentism and the culture wars from a

perspective and theoretical framework that are also of great

importance today.

The paper deals with the study of the ambivalence that characterizes

the modern world, in which the opening of possibilities of choice

that secular life entails has, as a counterpart, the danger of a

closure with respect to transcendent references (a closure that today

would be represented by "neo-immanentism" that points towards a

determinism that denies freedom). The article that we are considering

insists, therefore, and I consider that in a correct way, on the

study of the process of secularization of Christianity within which

the neo-immanentist tendency of techno-functionalism has been

constituted.

We could say that this techno-functionalist consciousness forgets its

own history, its own genesis: it forgets that the secularization of

the context does not necessarily demand the secularization of the

text, nor does it oblige the denial of transcendence. This negation

is a possibility, not a necessity or obligation. In this second case

there would be a hidden reabsolutization or a (negative)

resacralization of immanence, which thus loses its liberating

potential and becomes a new fatality. The danger would consist in the

pretension of suppressing the tension between the religious and the

secular in favor of the latter.

Well, I consider that this central thesis is developed and argued in

the article with solvency, rigor and precision, relying on abundant

classic studies of sociology, but also on the most recent

sociological bibliography.

We will only point out, as a contribution from our own field of work (philosophy), that such a thesis could also find support in philosophies such as that of E. Cassirer, who presents the symbolic way of being of the human being (and, therefore, also of all his cultural creations: myths, religions, arts, languages, sciences and techniques) or the Hermeneutics of Heidegger and Gadamer, according to which interpretation is constitutive of the way of being (hermeneutic) of the human being that, as it is effectively stated in the article, is distended between the realm of truth and that of meaning. Both philosophies come to recall, pointing in the same direction as this article, that consciousness (in this case techno-functionalist consciousness) does not sustain or generate itself but arises within processes that are not strictly conscious.

Author Response

Dear, reviewer and editor

We would like to thank the reviewer for his thorough reading and recommendations to improve the text. We would like to point out that the additions or corrections related to reviewer 2 are those that we have highlighted in the text in green color.

Improvements according to the reviewer's suggestions:

  1. This thesis could also find support in philosophies such as that of E. Cassirer or the Hermeneutics of Heidegger and Gadamer: We have introduced a footnote (number 2, page 5) in which we state the suggestion made by reviewer 2.

If the reviewer needs further clarification, we will be happy for answering.

Kind regards,

Reviewer 3 Report

This is a fine theoretical paper, very well anchored in current scholarship and contemporary debates, written in an elevated, attractive and scientifically rigorous language.  It is discussing highly actual problems and aims to respond to current pressing social, political and intellectual challenges. The structure of the paper is adequate and its argumentation well constructed, innovative and compelling. I highly recommend it for publication. 

The only suggestion I would like to make is that the Authors incorporate more references to local (regional, local, glocal) contexts, if they think fit. With the exception of the examples provided in lines 623-639 (China, Russia), the whole discussion is at global level,  with implicit references  to the Western context only (made explicit in lines 594-609). Perhaps this was the deliberate option of the authors in order to strengthen the central argument and avoid to go beyond the scope of the paper, so, by making my suggestion,  I do not insist that the authors make any change.  I consider the article suitable for publication in its present form. 

Author Response

First of all, we would like to thank the reviewer for his thorough reading of the paper and his kindness in his comments. In the text, responses to this reviewer are in brown color.

The only suggestion I would like to make is that the Authors incorporate more references to local (regional, local, glocal) contexts, if they think fit. With the exception of the examples provided in lines 623-639 (China, Russia), the whole discussion is at global level, with implicit references to the Western context only (made explicit in lines 594-609). Perhaps this was the deliberate option of the authors in order to strengthen the central argument and avoid to go beyond the scope of the paper, so, by making my suggestion, I do not insist that the authors make any change. I consider the article suitable for publication in its present form.

Answer: We have introduced an example as a footnote (footnote 5) to answer the question kindly introduced by the reviewer.

Reviewer 4 Report

The authors discuss the complex social processes that have led to the consolidation of a technocratic secular order, its relationship with neo-immanentist tendency and techno-functionalism and the conditions of possibility for other social sensibilities co-existing with the secular and technocratic.

The argument presented in the paper is well developed, different analytical propositions are referenced, however, there is some scope for improvement.

1.       A more clear explanation of the relationship between different key terms would make the text easier to follow. For example, terms such as secularism, technocratism, techninicism, datism all appear in the text repeatedly, but the relationships between them can only be deduced from the text and appear to be unstable. 

2.       It is unclear how the authors understand “three episodes that explain the hegemonic crystallization of techno-scientific expansion in the course of modernity”. Are they really explanation of this expansion? If so, in what sense? Are they “causes”? Is it possible that these episodes are parallel processes driven by all together different forces?

3.       The text requires some linguistic editing. In various place verb forms are incorrect (for example: in the abstract there should be “focuses” instead of “focus; in  l.183 there should be “takes”, instead of “take”; in l. 486 “While In modernity…” – why is In capitalized?; in l. 233 both dash “-“ and coma are used in the same place). The need for these type of corrections goes beyond the examples given.

4.       The language used by the authors gives agency to ideas, abstraction or worldview positions. It even ascribes these phenomena with feeling. A good example of this is the  following paragraph starting in line 611: “Techno functional Neo-immanentism has gained strength in the contemporary imaginary in an attempt to exhaust the semantic field of social interaction. It has not succeeded. There  are different cultural positions that feel challenged by dimensions that go beyond the  contours of empirical immediacy…” Personally I find this confusing, this stylistic choice makes the text more difficult to  follow.

5.       The following sentence in the abstract makes no sense: “the paper closes with a conclusion that takes up the path taken”. Also the sentence in lines 388-389 starting with “The amalgamated facts….” does not make sense to me.

6.       In the section 4 of the paper, it is not clear how Casanova’s argument about two possible developments of secularization related to the issues  of technologism and datism discussed earlier in the same section.

7.       In the conclusions the authors introduce the idea of emotivism somewhat unexpectedly. In my opinion it does not work well to tie up the  discussion presented in the main body of the paper and adds confusion.

I hope these remarks will help to improve the manuscript and I am looking forward to seeing it published.

Included in the comments above.

Author Response

First of all, we would like to thank the reviewer for his thorough reading of the paper and his kindness in his comments. In the text, responses to this reviewer are in blue color.

  1. A more clear explanation of the relationship between different key terms would make the text easier to follow. For example, terms such as secularism, technocratism, techninicism, datism all appear in the text repeatedly, but the relationships between them can only be deduced from the text and appear to be unstable.

Answer: We have introduced a paragraph in the introduction (lines 106-11) in which we establish a connection between the different main terms used in the work. The line of argument is established considering secularization as a context of action, the techno-functional neo-immanentism as one of the -secular- ways of articulating a religious sense in today's societies, being dataism or the imperative of the algorithm some of its main manifestations.

  1. It is unclear how the authors understand “three episodes that explain the hegemonic crystallization of techno-scientific expansion in the course of modernity”. Are they really explanation of this expansion? If so, in what sense? Are they “causes”? Is it possible that these episodes are parallel processes driven by all together different forces?

Answer: Thank you to the reviewer for highlighting this issue. We are indeed referring to three causes acting in parallel. We have modified this expression on page 4, paragraph 3, line 156.

  1. The text requires some linguistic editing. In various place verb forms are incorrect (for example: in the abstract there should be “focuses” instead of “focus; in l.183 there should be “takes”, instead of “take”; in l. 486 “While In modernity...” – why is In capitalized?; in l. 233 both dash “-“ and coma are used in the same place). The need for these type of corrections goes beyond the examples given.

Answer: Thanks to the reviewer for pointing out these typos. We have corrected all the ones he pointed out and have made a thorough reading of the text, identifying some more.

  1. The language used by the authors gives agency to ideas, abstraction or worldview positions. It even ascribes these phenomena with feeling. A good example of this is the following paragraph starting in line 611: “Techno functional Neo-immanentism has gained strength in the contemporary imaginary in an attempt to exhaust the semantic field of social interaction. It has not succeeded. There are different cultural positions that feel challenged by dimensions that go beyond the contours of empirical immediacy...” Personally I find this confusing, this stylistic choice makes the text more difficult to follow.

Answer: We have substituted the previous phrasing for that which appears between lines 657-661.  We believe that, in this way, it is much clearer for the reader.

  1. The following sentence in the abstract makes no sense: “the paper closes with a conclusion that takes up the path taken”. Also the sentence in lines 388-389 starting with “The amalgamated facts....” does not make sense to me.

Answer: We have modified the phrasing in the terms shown in the text in blue to make it easier to understand. With the same aim, we have deleted the term amalgamated.

  1. In the section 4 of the paper, it is not clear how Casanova’s argument about two possible developments of secularization related to the issues of technologism and datism discussed earlier in the same section.

Answer: To resolve this issue we have introduced some explanatory sentences between lines 516-519.

  1. In the conclusions the authors introduce the idea of emotivism somewhat
    unexpectedly. In my opinion it does not work well to tie up the discussion presented in the main body of the paper and adds confusion.

Answer: We have added a paragraph in the conclusion, specifically between lines 752 and 759, through which we focus the conclusion of the work on what has actually been worked on and not so much on issues to be addressed in future research

Reviewer 5 Report

The work meets all the conditions to be published.

Author Response

Dear, editor and reviewer

We would like to sincerely thank reviewer 3 for the complimentary words about our work and point out that we have not made any improvements because the reviewer did not ask us to do so. However, we upload the paper with the modifications made for the other reviewers.

Kind regards,

Reviewer 6 Report

This paper presents a reflection on a social fact at the center of social science debates. It analyzes with success, which is why it is considered suitable for publication in Religions, the emergence of what Philip Gorski calls neo-immanentism from its religious origins in the Judeo-Christian tradition to become a global phenomenon in our time. The authors use the concept correctly by seeing in ita cultural struggle between the secular and the religious, and the victory of the secular today in the form of a technological device, conceived as the great promise of man's liberation and control over reality. The text is clearly structured, its reading is comprehensible and its internal development offers unity and coherence.

From the thematic point of view, it is a contribution contribution to its field, based on sociological reflection on the religious and cultural fact and on the position of the religious narrative in times of technocratic predominance.technocratic predominance. However, we recommend the introduction of some theoretical and bibliographical references: First, the work  of the Frankfurt School thinkers, especially the Dialectic of Enlightenment by Adorno and Horkheimer, should be developed in the text since it is the source of inspiration for the great sociological analyses of the technocratic drive of modernity; Second, the authors must make reference to the reflection of Luc Boltanski and Laurent Thévenot in De la justification, les économies de la grandeur because this text have an impact on alternative forms to the domination of utilitarian codes and in the consolidation of agreements and consensus in today's globalized context; Finally, it might be of interest to develop further Steffen Mau's argument in his book The Metric Society when he refers to a new global context in which technology is incorporated into human life as a new form of functional intimacy and as a source of planetary quantitative comparison.

Author Response

Dear, reviewer and editor

We would like to thank the reviewer for his thorough reading and recommendations to improve the text. We would like to point out that the additions or corrections related to reviewer 4 are those that we have highlighted in the text in brown color.

Improvements according to the reviewer's suggestions:

  1. We recommend the introduction of some theoretical and bibliographical references: First, the work  of the Frankfurt School thinkers, especially the Dialectic of Enlightenment by Adorno and Horkheimer. With respect to this issue we have introduced a paragraph on page 5, lines 181-188 and a little addition on page 12, line 458.
  2. Second, the authors must make reference to the reflection of Luc Boltanski and Laurent Thévenot in De la justification, les économies de la grandeur. In this regard, we have introduced a footnote (number 6) on page 15.
  3. Finally, it might be of interest to develop further Steffen Mau's argument in his book The Metric Society. With respect to this issue we have introduced a quote on page 6, lines 206-216, a footnote (number 3) on page 6 itself, and a smaller addition on page 12, lines 456-458.

If the reviewer needs further clarification, we will be happy for answering.

Kind regards,